# Transcriptome Analysis Reveals Association of E-Class AmMADS-Box Genes with Petal Malformation in *Antirrhinum majus* L.

**DOI:** 10.3390/ijms26094450

**Published:** 2025-05-07

**Authors:** Dongmei Yang, Yiwen Chen, Yutong He, Jiayi Song, Ye Jiang, Meiyun Yang, Xingyan Zheng, Li Wang, Huizhen Hu

**Affiliations:** Yunnan Province Engineering Research Center for Functional Flower Resources and Industrialization, College of Landscape Architecture and Horticulture Sciences, Southwest Forestry University, Kunming 650224, China; donna_0507@swfu.edu.cn (D.Y.); chenyiwen17@swfu.edu.cn (Y.C.); hyt2022@swfu.edu.cn (Y.H.); siklib-91@swfu.edu.cn (J.S.); 15272180019@swfu.edu.cn (Y.J.); 3615lyr@swfu.edu.cn (M.Y.); xingyanzheng@swfu.edu.cn (X.Z.); wangli@swfu.edu.cn (L.W.)

**Keywords:** ABCDE model, hormone crosstalk, MIKC-type MADS-box genes, petal malformation, RNA-seq, Snapdragon (*A. majus* L.), WGCNA, VIGS

## Abstract

Snapdragon (*Antirrhinum majus*) serves as a model system for dissecting floral morphogenesis mechanisms. Petal malformation in *A. majus* impacts ornamental value, but its genetic basis remains poorly understood. We compared transcriptomes of the wild-type (Am11) and a petal-malformed mutant (AmDP2) to identify 2303 differentially expressed genes (DEGs), including E-class MIKC-type MADS-box genes *SEP3* (*AmMADS25/61/20/26*) and *SEP2* (*AmMADS85*). Weighted gene co-expression network (WGCNA), protein-protein interaction (PPI), qRT-PCR and virus-induced gene silencing (VIGS) analyses revealed interactions between SEP2/SEP3 and C/A/B-class *MADS-box* genes (*AG*, *AP1*, *AP3*), co-regulated MADS transcription factors (MTFs) AGL15 (AmMADS16), and auxin signaling genes (*SAUR1*, *IAA13*). qRT-PCR validated upregulation of *SEP3* and downregulation of *SEP2* in AmDP2. Our results suggest that E-class MADS-box genes are associated with petal malformation through coordinated interactions with hormonal pathways. These findings provide candidate targets for further functional studies in snapdragon.

## 1. Introduction

Snapdragon (*A. majus* L.), a perennial herbaceous plant of the Plantaginaceae family, is renowned for its unique floral architecture, diverse color phenotypes, and prolonged blooming period. As both a prominent ornamental species and a classical model organism for studying floral organ development [1,2,3], this dicotyledonous plant provides critical genetic resources for horticultural improvement due to its remarkable floral diversity.

Floral organs, a pivotal evolutionary innovation in angiosperms [4], arise from precisely regulated molecular networks [5,6]. Pioneering studies in *Arabidopsis thaliana* and *A. majus* established the foundational genetic framework for floral organ identity determination, evolving from the ABC model [7] to the comprehensive ABCDE model that integrates ovule development and organ boundary regulation [8,9,10,11]. This model specifies floral organ identities through combinatorial interactions of five gene classes [5,12,13]: The A-class (*APETALA1* (*AP1*), *AP2*, and *FRUITFUL*L (*FUL*)) + E-class (*SEPALLATA1* (*SEP1*), *SEP2*, *SEP3*, and *SEP4*) module directs sepal formation; A + B-class (*AP3*, *PISTILLATA* (*PI*)) + E-class genes determine petal development; and C-class (*AGAMOUS* (*AG*)) + E-class genes regulate pistil differentiation [9]. Notably, these genes maintain functional equilibrium through dynamic antagonistic-synergistic interactions. For instance, B-class *AP3/PI* suppresses C-class *AG* expression to prevent pistil overdevelopment [14], while E-class genes stabilize MADS-box protein complexes as core factor [15,16,17]. The universality of this regulatory paradigm has been validated across diverse plant species: In *Prunus mume*, *PmSEP1/4* regulates sepal differentiation, whereas *PmSEP2/3* governs petal-stamen-pistil development [18]; in *Pericallis hybrida*, ScAG interacts with ScAGL11 to suppress anthocyanin synthesis in ray florets [19]. These findings collectively highlight that spatiotemporal balance in ABCDE model genes module expression underpins the molecular evolution of floral morphological diversity.

Central to this regulatory network are MTFs, which exhibit conserved functionality across eukaryotes [20,21,22,23,24]. Structurally, most ABCDE model genes (excluding *AP2*) encode MIKC-type proteins [25], mediating floral organ identity via homo-/heterodimerization [26,27]. For example, *TaAGL6* in wheat promotes stamen development by activating *TaAP3* [28], while MTFs members SHORT VEGETATIVE PHASE (SVP) and SUPPRESSOR OF OVEREXPRESSION OF CONSTANS 1 (SOC1) in persimmon interacts with the sex-determinant MeGI to regulate pistil differentiation [29]. However, floral morphogenesis extends beyond MADS-box genes, involving synergistic regulation by other TF families, including MYB [30,31,32,33], KNOX [34], and bHLH [35], as well as hormonal crosstalk, suggesting that floral organ morphogenesis is driven by multidimensional regulatory networks. 

Despite the fact that progress has been made, critical questions remain in snapdragon biology: How do *MADS-box* genes coordinate with hormonal signals to maintain floral development? What molecular mechanisms involved in petal malformation in mutants like AmDP2 (distinctive and penetrant petal malformations, including fused petals and disrupted organ boundaries)? To address these, we compared transcriptomes of wild-type (Am11) and special malformed mutant (AmDP2) petals, integrating WGCNA, PPI, qRT-PCR, and VIGS analyses. These findings suggest potential roles of a gene-TF-hormone regulatory framework where antagonistic E-class genes (SEP3 vs. SEP2) modulate floral development through coordinated interactions with hormonal pathways. This model advances our understanding of floral morphogenesis and identifies potential targets for genetic improvement in ornamental plants.

## 2. Results

### 2.1. Sequencing Read Filtering and Identification of Differentially Expressed Genes (DEGs)

Phenotypic analysis revealed distinct petal malformation phenotypes in the *A. majus* inbred line Am11, designated as AmDP2, while control plants maintained normal flower morphology (Figure 1A). The transcriptomic profiling of six cDNA libraries (three biological replicates per genotype) generated a total of 38.34 Gb of raw data using the DNBSEQ platform (BioProject ID: PRJNA1238210). Quality assessment demonstrated high data integrity, with Q20/Q30 values exceeding 98%/92% and GC content stabilized between 44.43 and 44.79 (Appendix A). The fragments per kilobase of transcript per million mapped reads (FPKM)-based expression quantification and visualization via violin plots and stacked bar charts confirmed even read distribution and reproducibility across the samples (Figure 1B,C), validating suitability for differential gene analysis.

Systematic comparison identified 29,993 expressed genes, with 88.2% (26,460 genes) showing co-expression in both lines (Appendix A). DEGs were identified using edgeR v3.36.0 with the following thresholds: |log_2_(fold change)| ≥ 0.5 and significant false discovery rate (FDR) < 0.05, yielding 2303 DEGs (Figure 1D). Of these, 1051 genes were upregulated and 1252 downregulated in AmDP2 (Figure 1D). Cluster analysis revealed distinct expression modules (Appendix A), suggesting regulatory network involvement in specific biological pathways.

### 2.2. GO and KEGG Pathway Enrichment of DEGs

Gene Ontology (GO) annotation categorized the DEGs into three primary categories: biological process (BP, 54.3%), cellular component (CC, 28.1%), and molecular function (MF, 17.6%) (Figure 2A). The top 20 enriched GO terms were dominated by membrane-associated components (GO:0016020, GO:0031224, GO:0016021), with additional representation from cytoskeletal proteins (GO:0005856, GO:0015629) and transporter activities (GO:0005215, GO:0022857) (Figure 2B). Notably, membrane integral components (34.6% of total GO terms) highlighted the potential role of transmembrane regulation in petal malformation.

The Kyoto Encyclopedia of Genes and Genomes (KEGG) pathway analysis grouped DEGs into five core functional modules: metabolism (61.28%, 1282/2092 genes), genetic information processing, environmental information processing, cellular processes, and organismal systems (Figure 2C). Sub-pathway analysis revealed carbohydrate metabolism (256 genes) and signal transduction (139 genes) as significantly enriched categories. Mechanistic studies identified three key pathways: starch and sucrose metabolism (KEGG: ko00500), MAPK signaling (KEGG: ko04016), and clathrin-mediated endocytosis (KEGG: ko04144) (Figure 2D), suggesting metabolic imbalance, disrupted MAPK signaling cascades, and impaired membrane trafficking during petal development.

Furthermore, KEGG annotation uncovered multiple MADS-box genes related to floral development, including *SVP* (*Am08g16940*) and *SOCI* (*Am05g19210*, *Am02g37900*), which were enriched in Metabolism and Cellular Processes. Their involvement in regulatory networks underscores the combinatorial roles of transcription factors and metabolic pathways in petal morphogenesis.

### 2.3. Identification and Evolutionary Conservation of MIKC-Type AmMADS Genes

Using the ABCDE floral development model (Figure 3A), we systematically identified 86 MIKC-type MADS-box members in *A. majus* homology searches against 45 *Arabidopsis* MIKC-type MADS-box protein sequences using HMMER v3.0 and BLAST (v.0.6735) algorithms (Appendix A). These genes, named *AmMADS1*-*AmMADS86* were annotated based on chromosomal localization and structural domains (SRF/M-type and K-box/MIKC-type) (Appendix A). Phylogenetic analysis via MEGA v11.0 revealed 12 evolutionarily conserved subfamilies, including MIKC*, SEP, AGL6-like, AP1, SOC1, AGL15-like, AGL12-like, ANR1, SVP, PI, TT16, and AG (Figure 3B), confirming dicotyledonous conservation. Functional classification demonstrated that 21 ABCDE model genes were identified, with E-class genes dominating (6 genes: *AmMADS67/61/25/26/85/20*), followed by B-class (5 genes: *AmMADS63/62/34/37/3*), A-class (5 genes: *AmMADS33/43/21/66/86*), and C/D-class (5 genes: *AmMADS18/49/2/40/6*). Notably, no FLC subfamily members were detected.

The RNA-seq analyses identified differential expression patterns in 18 ABCDE model genes and 16 MTFs (Figure 3C,D). Specifically, E-class genes (*AmMADS67/61/25*), B-class genes (*AmMADS37/3/34*), A-class genes (*AmMADS21/66/86*), and C/D-class genes (*AmMADS2/40/6*) exhibited significant upregulation in the petals of the heteromorphic mutant AmDP2, whereas E-class genes (*AmMADS85/20*), A-class genes (*AmMADS33/43*), and C/D-class gene *AmMADS18* showed marked downregulation. These expression patterns suggest their critical roles in petal malformation.

Protein-protein interactions (PPIs) are critical for plant developmental regulation [36]. By constructing a homologous *Arabidopsis* PPI network through STRING v11.5 (confidence >0.7) [37], we dissected ABCDE model gene interactions in *A. majus*. The network was predominantly localized to nucleus/cytoplasm compartments (Appendix A). Key regulatory interactions emerged between E-class SEP3 (AmMADS25/61/20/26), SEP2 (AmMADS85), and SEP1 (AmMADS67) as core factors, which interacted with A/B/C/D-class genes and MTFs. Specifically, SEP3 (AmMADS25/61/20/26) formed core hubs through autointeractions and interactions with B-class AP3 (AmMADS37/3) and PI (AmMADS34), A-class AP1 (AmMADS33/43/21/86), C/D-class AG (AmMADS6), STK (AmMADS2/40), and SHP1 (AmMADS18), and MTFs member AGL15 (AmMADS16). Additionally, B-class AP3 (AmMADS37/3) and A-class AP1 (AmMADS33/43/21/86) interacted with MTFs members AGL13 (AmMADS12), SOC1 (AmMADS13/46), and SVP (AmMADS84) (Figure 3E).

### 2.4. Cis-Acting Element and TFBS Analysis in ABCDE Model DEGs Promoters

To elucidate the regulatory networks governing petal morphogenesis, we performed cis-acting element and transcription factor binding site (TFBS) analyses on the promoters of 18 differentially expressed ABCDE model genes. PlantCARE-based scanning revealed significant enrichment of growth-related hormone response elements for brassinosteroid (BR), gibberellin (GA), and auxin (IAA), with E-class genes exhibiting the most extensive distribution (Figure 4A). Notably, stress-related hormone response elements for methyl jasmonate (MeJA) and ethylene (ET) were also enriched, suggesting a potential balance between growth promotion and stress resistance mechanisms in petal development.

TFBS analysis identified abundant binding motifs for MYB, Dof, and C2H2-type transcription factors (TFs), with the notable enrichment of MIKC_MADS, NAC, and WRKY-type sites (Figure 4B). Quantitative analysis revealed that E-class *SEP3* (*AmMADS25/61*), *SEP2* (*AmMADS85*), and *SEP1* (*AmMADS67*) exhibited the highest TFBS density, followed by B-class *AP3* (*AmMADS37*) and C/D-class *SHP1* (*AmMADS18*). This pattern highlights their potential roles as core regulatory hubs integrating multi-signal pathways, including hormonal crosstalk and stress responses.

### 2.5. Co-Expression Networks Among Phytohormones, TFs, and ABCDE Model DEGs

To dissect the regulatory networks governing petal malformation in *A. majus*, we performed weighted gene co-expression network (WGCNA) analysis on 2303 DEGs (Appendix A). The analysis identified 11 co-expression modules with a soft-threshold of β =15 and R^2^ = 0.8. Module-trait association revealed extreme positive correlation (r = 1, *p* < 0.001) between the MEgreen module and AmDP2 phenotypes, while the MEblue module showed significant negative correlation (r = −0.92, *p* = 0.01) (Figure 5A), suggesting bidirectional regulatory roles in petal development. These modules were prioritized for downstream analysis (Appendix A).

Phytohormones function through synergistic crosstalk rather than independent action. Based on hormone-responsive element enrichment in ABCDE model gene promoters (Figure 4), we constructed co-expression networks between differentially expressed ABCDE model genes and phytohormone pathways in the MEblue and MEgreen modules (Figure 5B and Appendix A). The analysis revealed significant co-expression between E/B/A-class genes and hormone synthesis/signaling genes. Notably, E-class genes *SEP3* (*AmMADS25/61*) and *SEP2* (*AmMADS85*), B-class genes *AP3* (*AmMADS3/37*) and *PI* (*AmADS34*), and A-class gene *AP1* (*AmMADS33/43*) exhibited strong correlations with hormones in these modules (Figure 5C). Detailed expression profiling showed that *SEP3* (*AmMADS25/61*) displayed concordant expression patterns with *AP3* (*AmMADS3/37*) and *PI* (*AmADS34*). Their expression levels were significantly positively correlated with brassinosteroid synthases (*CYP85A1/CYP77A2*), ethylene response factor *ERF3*, gibberellin receptor *GID1*, and auxin-related gene *SAUR64* (Figure 5C) while negatively correlated with auxin-responsive genes (*SAUR1/IAA13*), brassinosteroid receptor *BR*, ethylene-responsive gene *EREBP*, and gibberellin receptor *GID1-1*. In contrast, the co-expression patterns of A-class *AP1* (*AmMADS33/43*) and E-class *SEP2* (*AmMADS85*) were inversely regulated, suggesting antagonistic roles in hormonal crosstalk. These findings suggest that E/B/A-class genes regulate petal malformation through dynamic crosstalk with phytohormone signaling pathways and mutual interactions among themselves, aligning with the results observed in Figure 3E.

Furthermore, we explored TFs interactions in the MEgreen (Appendix A) and MEblue (Appendix A) modules. Heatmap analysis (Figure 5D,E) revealed exclusive co-expression between E-class genes and TFs. Network analysis of six E-class genes (*AmMADS85/67/26/61/25/20*) identified *SEP3* (*AmMADS25/61*) and *SEP2* (*AmMADS85*) as hub nodes (Figure 5F). Notably, these paralogous E-class genes exhibited antagonistic expression patterns: *SEP2* (*AmMADS85*) was highly expressed in Am11 and co-expressed with *MYB25*, *MLP*, WRKY, and *bHLH51* genes, whereas *SEP3* (*AmMADS25/61*) was enriched in AmDP2 and synchronized with *DPR1*, *CRK25*, *MYB1*, *SASPL*, *KAT3*, and *SUT1* genes. This functional specialization through divergent regulatory networks highlights their potential as master regulators of petal morphogenesis.

### 2.6. Validation of Key ABCDE Model Genes by qRT-PCR

To experimentally validate the roles of candidate ABCDE model genes identified in RNA-seq analyses, we performed qRT-PCR on nine core genes: *AmMADS25/61/85* (E-class), *AmMADS37/3/34* (B-class), and *AmMADS43/33/21* (A-class) (Figure 6). RNA-seq and qRT-PCR data showed high consistency for seven genes, excluding *AmMADS34* and *AmMADS21*. E-class genes *AmMADS25/61* (*SEP3*) and B-class gene *AmMADS37/3* (*AP3*) exhibited significant upregulation in the petals of *AmDP2* mutants (fold changes: 10.20–100.42), whereas E-class gene *AmMADS85* (*SEP2*) and A-class genes *AmMADS43/33* (*AP1*) were preferentially expressed in wild-type Am11 (fold changes: 15.33–94.51). Notably, E-class genes displayed antagonistic expression patterns: *AmMADS25/61* (*SEP3*) was upregulated in AmDP2, while *AmMADS85* (*SEP2*) showed converse expression in Am11. These differential expression profiles confirmed their critical roles in mediating petal malformation through competing regulatory mechanisms.

### 2.7. Silencing of AmSEP2 Promotes Petal Malformation in A. majus: Gene Network Interactions

To elucidate the functional interplay between core *ABCDE* model genes, co-expressed phytohormones, TFs, and MTFs, we transiently silenced *AmMADS85* (*SEP2*, E-class), a key candidate gene showing significantly reduced expression in petal-malformed AmDP2 mutants, in petal-malformed AmDP2 mutants, via agroinfiltration of the pTRV2-*AmSEP2* vector in wild-type *A. majus* Am11 petals. qRT-PCR confirmed efficient gene silencing: *AmSEP2* transcript levels in pTRV2-*AmSEP2*-inoculated petals decreased by 46.17%, 54.79%, and 70.32% at 6, 12, and 18 h post-inoculation (hpi), respectively, compared to the empty-vector control (pTRV1+pTRV2-empty) (Figure 7A), demonstrating successful suppression via the TRV-VIGS system.

Given the putative direct interaction between AmSEP2 and E-class SEP3 (AmMADS25/61) as well as C/D-class AG (AmMADS6), we quantified their transcript expression in petals at 18 hpi. *SEP3* and *AG* exhibited compensatory upregulation by 70.52% (Figure 7E)/96.07% (Figure 7F), and 378.47% (Figure 7B) in *AmSEP2*-silenced petals compared to the controls, suggesting the activation of downstream MADS-box TFs. Further analysis of *SEP3*-interacting partners included A-class *AP1* (*AmMADS43*), B-class *AP3* (*AmMADS37*), MTFs *AGL5* (*AmMADS16*), and hormone-responsive genes *AmSAUR1/AmIAA13*. We found that *AP1* expression decreased by 90.92% in *AmSEP2*-silenced petals compared to the controls (Figure 7C), while the *AP3* level were unchanged (Figure 7D). Notably, *AmSAUR1/AmIAA13*, which show a negative correlation with *AmSEP3*, displayed reduced expression by 55.41% (Figure 7H) and 61.89% (Figure 7I), respectively.

These findings collectively indicate that *AmSEP2* silencing promotes petal malformation may primarily through the activation of the E-class SEP2/SEP3-AGL15-SAUR1/IAA13 regulatory axis, preliminarily establishing a three-dimensional network integrating ABCDE model genes, MTFs, and phytohormone signaling pathways (Figure 8). However, the functional relevance of these interactions requires experimental validation through yeast two-hybrid (Y2H), co-immunoprecipitation (Co-IP), and gene-editing approaches to confirm their direct roles in snapdragon petal malformation.

## 3. Discussion

### 3.1. Integrated Complex Approaches as a Powerful Strategy for Functional Gene Discovery

Transcriptome sequencing has emerged as a high-throughput and efficient methodology for functional gene discovery and pathway analysis in horticultural research. Notable studies in *Pogostemon cablin* [38], *Petunia hybrida* [39], and *Dianthus caryophyllus* [40] have successfully elucidated floral development mechanisms. However, the sole reliance on RNA-seq for functional gene discovery has yielded limited success. Meanwhile, despite the success of the ABCDE model in *A. majus*, petal malformation remains an understudied phenomenon with scarce mutant materials available. To address this gap, we employed a unique AmDP2 mutant exhibiting petal malformation in *A. majus*—a well-established model organism for flower development studies. We developed an integrated complex analytical framework combining phenotypic characterization, RNA-seq, genome-wide analysis, WGCNA, PPI prediction, and experimental validation via qRT-PCR and VIGS. This comprehensive approach enabled the systematic delineation of the molecular regulatory networks governing petal malformation in *A. majus* AmDP2.

Initial analysis of ABCDE model genes in petal malformation: to investigate whether petal malformation in AmDP2 is driven by ABCDE model genes, we performed global differential expression analysis of DEGs. Notably, 18 out of 21 ABCDE model genes were identified as DEGs (Figure 3C), suggesting their pivotal roles in floral malformation. The subsequent genome-wide identification of MIKC-type MADS-box genes in *A. majus* revealed 86 members, including 21 canonical ABCDE model genes and 65 MTFs belonging to 12 evolutionarily conserved subfamilies, such as SOC1, AGL15-like, AGL12-like, and SVP, etc. Among these, 16 MTFs exhibited differential expression (Figure 3D), highlighting their potential regulatory significance in petal malformation. 

Protein-protein interaction network analysis of DEGs belonging to the MADS - box gene family: Using computational PPI prediction algorithms informed by *Arabidopsis* studies, we identified key regulatory interactions among the 18 DEGs and 16 MTFs. Strikingly, we uncovered scaffold functions for E-class members *SEP3* (*AmMADS25/61/20/26*), *SEP2 (AmMADS85*), and *SEP1* (*AmMADS67*), which interacted with A/B/C/D-class members and other MTFs (Figure 3E). This finding corroborates previous research, which suggests that these genes maintain functional equilibrium through dynamic antagonistic-synergistic interactions [14,15,16]. This network reduction significantly narrowed candidate genes for further investigation. 

Numerous studies have reported that upstream TFs play pivotal roles in regulating gene expression, with cis-acting elements in promoter regions serving as critical regulatory determinants [41]. Moreover, previous research has found that both hormonal signal transduction pathways and transcriptional regulation can modulate the process of flower development [30,31,32,33,34,35,42,43,44,45]. In this study, we further analyzed the cis-acting elements in the promoter regions of 18 differentially expressed genes in the ABCDE model. We found that these genes, especially those in the E-class family such as *SEP3* (*AmMADS25/61*), *SEP2* (*AmMADS85*), and *SEP1* (*AmMADS67*), contain a large number of elements related to growth-related hormonal signal pathways and TFBS (Figure 4). This indicates that the E-class MIKC-type MADS-box genes, which drive petal malformation, are involved in a complex molecular network regulation during their function.

WGCNA-based co-expression module identification: WGCNA analysis of 2303 DEGs identified two modules with significant trait associations: MEgreen (r = 1, *p* < 0.001) showed an extreme positive correlation with AmDP2 phenotypes, while MEblue (r = −0.92, *p* = 0.01) exhibited a negative correlation (Figure 5A). Ultimately, a significant co-expression was discovered between E/B/A-class genes and hormone synthesis/signaling genes, such as *SAUR1* and *IAA13* (Figure 5B,C). Moreover, only three E-class genes, *SEP3* (*AmMADS25/61*) and *SEP2* (*AmMADS85*), acting as hub nodes, interacted with TFs like *MYB25*, *MLP*, *WRKY*, and *bHLH51* (Figure 5D–F). This further indicates from another dimension that these three E-class genes, *SEP3* (*AmMADS25/61*) and *SEP2* (*AmMADS85*), occupy a central and pivotal position in the complex network regulation driving petal malformation. 

Experimental validation and regulatory framework proposal: consistent with the RNA-seq, qRT-PCR and VIGS validation (Figure 6 and Figure 7), our integrated approach proposes a novel regulatory model where E-class MADS-box genes act as central factor coordinating with TFs and hormones. Specifically, we hypothesize that SEP2/*SEP3-AGL15-SAUR1/IAA13* crosstalk disrupts auxin signaling pathways, while *AP3-AGL13-SAUR1/IAA13* and *AP1-SVP/SOC1-SAUR1/IAA13* pathways exhibit antagonistic roles in floral development (Figure 8). This combinatorial regulatory framework not only provides potential mechanistic insights into petal malformation but also establishes a paradigm for complex data integration in functional gene discovery based on the basic data of DEGs.

### 3.2. E-Class AmMADS-Box Genes: Central Factor in Petal Malformation of A. majus

The classical ABCDE model specifies floral organ identities through combinatorial interactions among five gene classes [5,12,13]. The A-class (AP1, AP2, FUL) + E-class (SEP1-SEP4) module governs sepal formation, while A- + B-class (AP3, PI) + E-class genes determine petal development. C-class (AG) + E-class genes regulate pistil differentiation [9]. E-class genes function as core factor by stabilizing MADS-box protein complexes, a pivotal mechanism for floral organogenesis. The SEP subfamily plays essential roles in floral meristem determination and organ identity specification. SEPs exhibit organizational capacity through heterodimerization [16] and tetramerization [15], with recent studies demonstrating that SEP3 tetramers are required for floral meristem determinacy and second-to-fourth whorl organ identity [17]. These discoveries accentuate the fundamental significance of E-class AmMADS-box genes in angiosperm development. Nevertheless, their regulatory mechanisms remain incompletely elucidated, particularly in the context of petal malformation.

Through multiple integrative analysis, we identified E-class AmMADS-box genes *SEP3* (*AmMADS25/61*) and *SEP2* (*AmMADS85*) as dominant factors driving petal malformation in AmDP2, revealing species-specific functional divergence within the SEP subfamily compared to other angiosperms. Unlike *Arabidopsis* triple (*sep1 sep2 sep3*) and quadruple (*sep1 sep2 sep3 sep4*) mutants that exhibit homeotic conversions to sepaloid/leaf-like organs [46,47], *A. majus* SEP3/SEP2 dysregulation primarily disrupts petal morphology (Figure 6), with no sepaloid transformations, indicating the evolutionary neofunctionalization of SEP genes in petal patterning. This contrasts with *P. mume*, where *PmSEP1/4* regulate sepal differentiation while *PmSEP2/3* govern petal-stamen-pistil development [18], paralleling the antagonistic expression patterns of *AmSEP3/SEP2* in AmDP2 vs. wild-type petals (Figure 6 and Figure 7). Notably, *AmSEP3/SEP2* uniquely integrate hormonal crosstalk (e.g., *SEP3-AGL15-SAUR1/IAA13* axis) to drive malformation, a mechanism absent in *Prunus* SEP regulation. Comparative studies across angiosperms further reveal clade-specific functional innovations: in *P. hybrida*, SEP1/SEP2/SEP3 homologs (e.g., FBP2) collaborate with AGL6 to define sepal/petal identity, whereas FBP9 subclade members (e.g., FBP9/FBP23) regulate floral meristem determinacy [48]; however, *A. majus* SEP3/SEP2 lack orthologs in FBP9 clades but acquire novel interactions with MTFs (e.g., AGL15) and hormonal pathways, reflecting evolutionary adaptations distinct from ancestral sepal specification roles. These data underscore that SEP subfamily members underwent functional partitioning during angiosperm evolution, with AmSEP3/SEP2 evolving context-dependent regulatory mechanisms (e.g., hormonal feedback loops) to fine-tune petal morphogenesis, marking a significant divergence from conserved sepal–petal boundary roles in other species.

Additionally, our PPI and VIGS analysis revealed scaffold functions for E-class members interacting with A/B/C/D-class genes (Figure 3E and Figure 7). Notably, SEP3 (AmMADS25/61) formed complexes with B-class AP3 (AmMADS37/3) and PI (AmMADS34), A-class AP1 (AmMADS33/43/21/86), and C/D-class AG (AmMADS6), STK (AmMADS2/40), and SHP1 (AmMADS18). This pattern aligns with the established models of ABCDE gene interactions mediating floral organ identity through homo-/heterodimerization-driven equilibrium maintenance [26,27,28]. For instance, wheat *TaAGL6* promotes stamen development via the activation of B-class *TaAP3* [27], demonstrating conserved regulatory mechanisms across species.

Collectively, our findings establish E-class *AmMADS-box* genes as central factor in *A. majus* petal malformation. The dual role of *SEP3* and *SEP2* in both promoting malformation and exhibiting antagonistic expression patterns underscores their multifaceted regulatory capabilities. Furthermore, the identification of species-specific interaction networks (e.g., AmMADS25/61-AP3 vs. TaAGL6-TaAP3 interactions) implies evolutionary divergence in MADS-box gene function. These insights not only clarify the molecular basis of petal malformation in *A. majus* but also provide a framework for comparative studies of floral development mechanisms across angiosperm species. The functional diversity of these genes warrants in-depth investigation.

### 3.3. Complex Regulatory Networks Underlying Petal Malformation in A. majus

Based on revealing that E-class AmMADS-box genes serve as central factor in *A. majus* petal malformation, this study also preliminarily predicted their multiple regulatory networks. First, we characterized the interaction network between E-class AmMADS-box genes and B/A/C/D-class MADS-box genes, which forms the core module for floral organ specification. Second, the regulatory crosstalk between ABCDE model genes and MTFs was revealed, demonstrating hierarchical control over floral development. Third, we integrated the interaction among ABCDE model genes, MTFs, and hormonal signaling pathways to fine-tune petal morphogenesis. Specifically, the E-class *SEP2/SEP3-AGL15-SAUR1/IAA13* crosstalk disrupted the auxin signaling pathways, while the B-class *AP3-AGL13-SAUR1/IAA13* and A-class *AP1-SVP/SOC1-SAUR1/IAA13* pathways play antagonistic roles in petal malformation. This multidimensional regulatory network underlying petal malformation in *A. majus* not only provides a new perspective for understanding the mechanism of petal malformation and enriches the regulatory mechanism of flower development patterns but also raises new questions about the specificity of flower development among species.

Regarding the interaction between ABDCE model genes and MTFs, such as SEP3-AGL15, AP3-AGL13, and AP1-SVP/SOC1, previous studies mainly focused on SVP/SOC1. It was found that SVP/SOC1 in persimmon interacts with the sex-determinant MeGI to regulate pistil differentiation [29]. Among the common direct targets identified, AP2-like genes that repress FT and SOC1 expression were down-regulated by SOC1 but up-regulated by SVP, revealing complex feedback regulation among key genes for integrating flowering signals [49]. Some studies have also revealed the interaction between MTFs member AGL and ABDCE model genes [50]. For instance, AGL13 can interact with AG to form quartet-like complexes (AGL13-AG) and interact with AG-AP3-PI to form a higher-order heterotetrameric complex (AGL13-AG-AP3-PI) involved in pollen morphogenesis, anther cell layer formation, and ovule development. It is hypothesized that *AGL13* might be a putative ancestor for the E-functional genes specifying male and female gametophyte morphogenesis in plant evolution. The authors in [51,52] found that AGL15 accumulates at the shoot apex and the base of leaf petioles during the vegetative phase and later in floral buds during the reproductive phase. The constitutive or ectopic expression of AGL15 led to prolonged sepal and petal longevity and delayed flowering transition, floral organ abscission, and fruit maturation, suggesting its important role in floral organ development and longevity. Later, AGL15 and AGL18 were found to act together to repress the floral transition in *Arabidopsis* [53] by forming a complex that activates the expression of microRNA miR156 [54]. The most important interaction predicted in this study is *SEP2/SEP3-AGL15*, for which there is currently no similar report. Further research is needed to determine the nature of the interaction in this dimension and whether it truly regulates petal malformation.

There are few reports on the more complex interactions among ABDCE model genes, MTFs and hormone signaling pathways. Existing reports on flower development are mostly one-dimensional. For example, TFs regulate MADS-box genes. It has been reported that TCP TFs interact with MADS-box proteins to drive the evolution of complex inflorescence structures in *Gerbera hybrida* [45], and in rice (*Oryza sativa*), OsMADS3 coordinates with DROOPING LEAF and OsMADS13 to regulate floral organ identity and meristem determinacy [55]. Also, MADS-box factors are regulated by hormones. BRE encodes an AP2 required for floral organogenesis, especially carpel initiation, and acts through the auxin pathway in strawberry [56]. SOC1 is a key flowering promoter integrating signals from photoperiod, temperature, hormones, and age-related pathways [57]. It forms complexes with other factors to regulate flower-related processes and responds to multiple hormone signaling pathways and nutrient status. Current research mainly focuses on how hormones affect flower development. In *P. hybrida*, gibberellins (GA) enhance corolla anthocyanin biosynthesis by upregulating chalcone synthase (CHS) gene expression [58]. Silencing the auxin response factor OsARF1 in rice induces dwarfism, delayed flowering, reduced fertility, or complete floral absence [59]. In Rosa hybrida, the ethylene-responsive transcription factor RhEIN3 directly binds to the promoter of RhPMP1, promoting its transcript accumulation specifically at the adaxial base of petals [60]. The Exogenous application of cytokinin (CTK) analogs significantly induces pistil primordium development in *Plukenetia volubilis* [43]. Notably, phytohormones do not act in isolation but rather orchestrate floral organogenesis through intricate crosstalk. For instance, GA promotes petal expansion by degrading DELLA proteins to activate SEP3 expression [61], while auxin (IAA) regulates floral organ number via the ARF3-MADS axis [42]. In *Arabidopsis*, brassinosteroids (BRs) delay flowering through the ABI3-FLC pathway [62,63], with core transcription factors BZR1 and BIM1 directly activating FLC to suppress floral transition [64]. Cytokinin (CTK) modulates ovule development by enhancing SPL and AG expression, thereby regulating PIN1-mediated auxin transport [44].

Our findings bridge molecular genetics and systems biology, proposing a framework where hormonal signals (auxin) modulate MTFs (e.g., *AGL15*) to fine-tune petal patterning. This multiple perspective advances beyond classical models, offering testable hypotheses for floral evolution and plant improvement. Future research should prioritize the functional validation of candidate genes via CRISPR/Cas9-mediated genome editing, particularly for *AmMADS25/61/85* and *AmMADS16*, which are implicated in petal malformation. These functionally validated candidates could serve as molecular markers for marker-assisted selection (MAS), leveraging their validated roles in floral development to bypass the limitations of traditional phenotypic screening, such as environmental noise and Mendelian inheritance complexities [65,66]. Comparative genomic analyses with other *Antirrhinum* mutants are further warranted to resolve conserved vs. divergent regulatory modules underlying floral morphogenesis, thereby refining our understanding of evolutionary conservation in MADS-box gene networks.

While our study focuses on MADS-box members in primary metabolism (e.g., starch/sucrose metabolism, Figure 2D), emerging evidence suggests their potential role in secondary metabolite regulation—a largely unexplored advancement in *A. majus*. KEGG pathway analysis revealed that *SVP* (*Am08g16940*) and *SOCI* (*Am05g19210/Am02g37900*) are enriched in metabolic and cellular processes, hinting at possible crosstalk with secondary metabolite biosynthesis. For instance, the overexpression of *CsMADS3* in citrus and tomato enhances carotenoid biosynthesis while upregulating chlorophyll degradation genes [67], paralleling potential *Antirrhinum* MADS-box regulation of terpenoid/phenylpropanoid pathways via downstream enzymes (e.g., *CYP76B6*) [3]. Notably, the MAPK signaling pathway (KEGG: ko04016), significantly enriched here, is tightly linked to secondary metabolism. MAPK cascades regulate JA biosynthesis, a hormone critical for defense-related secondary metabolite accumulation [68]. This hypothesis is further supported by JA-responsive elements in E-class MADS-box genes (Figure 4A). Future studies integrating metabolomics and ChIP-seq could clarify whether *A. majus* MADS-box members directly govern secondary metabolite pathways, expanding their functional scope in floral development. 

## 4. Material and Methods

### 4.1. Plant Materials and Sample Preparation

This study utilized the inbred line *A. majus* Floral Showers Purple (Am11) and its petal malformation mutant (AmDP2), cultivated at the Arboretum Research Base of Southwest Forestry University (25°03′ N, 102°45′ E, Kunming, China). Plants were grown under standardized conditions (20–25 °C, natural light) with uniform agronomic practices. During the peak flowering period (June 2024), petals from Am11 (normal) and AmDP2 (malformed) were collected, immediately flash-frozen in liquid nitrogen, and stored at −80 °C for downstream analyses.

### 4.2. cDNA Library Construction and RNA-Seq

Fresh samples of Am11 and AmDP2 were sent to the DNBSEQ platform (BGI Co., Ltd., Shenzhen, China) for mRNA library construction and RNA sequencing. Three biological replicates were set for each sample. Total RNA was used to construct six cDNA libraries following the standard protocols. Specifically, total RNA was processed by either mRNA enrichment or rRNA depletion methods. For mRNA enrichment, magnetic beads with OligodT were used to enrich mRNAs with polyA tails. In the rRNA depletion approach, DNA probes were hybridized to rRNAs, followed by the selective digestion of DNA/RNA hybrid strands with RNaseH. Subsequently, the DNA probes were digested with DNaseI. After purification, the desired RNA was obtained. The obtained RNA was fragmented using fragmentation buffer. Reverse transcription was carried out with random N6 primers, and then the second-strand cDNA was synthesized to form double-stranded DNA. The ends of the synthesized double-stranded DNA were blunted and phosphorylated at the 5′ end, while a protruding “A” was created at the 3′ end to form a sticky end. Then, a bubble-shaped adapter with a protruding “T” at the 3′ end was ligated. The ligation products were amplified by PCR using specific primers. The PCR products were heat-denatured into single-strands, and then single-stranded DNA was circularized using a bridge primer to obtain a single-stranded circular DNA (cDNA) library. Finally, qualified RNAs were subjected to sequencing. A total of 38.34 Gb of raw data was generated by the DNBSEQ platform. After quality control (removing adapters, low-quality reads, and N content >1%), clean reads ranged from 42.09 M to 43.13 M per sample (Appendix A).

### 4.3. Sequencing Data Filtering and DEG Identification

Raw sequencing data were subjected to quality control using SOAPnuke (https://github.com/BGI-flexlab/SOAPnuke, accessed on 6 October 2024) for preprocessing. Adapter-contaminated reads, reads containing >1% undefined bases (N), and low-quality reads (defined as reads with >40% of bases having a Phred quality score <20) were rigorously filtered to obtain clean reads. Clean reads were aligned to the snapdragon reference genome [69] using HISAT2 (http://www.ccb.jhu.edu/software/hisat, accessed on 6 October 2024), followed by secondary alignment validation with Bowtie2 (https://bowtie-bio.sourceforge.net/bowtie2/index.shtml, accessed on 6 October 2024) [70]. Transcript expression quantification was performed via RSEM (https://github.com/deweylab/RSEM, accessed on 6 October 2024), with expression levels normalized as FPKM [71]. FPKM was used to quantify the expression levels of the samples, and visualization was performed using violin plots and stacked bar charts.

DEGs were identified using thresholds of |Log_2_(Fold Change)| ≥ 0.5 and *p*-value <0.05. Genes with Log_2_(AmDP2_FPKM/Am11_FPKM) > 0 were defined as upregulated, while those with Log_2_(AmDP2_FPKM/Am11_FPKM) < 0 were classified as downregulated. The hierarchical clustering analysis of DEG expression profiles was conducted using the pheatmap function within the RSEM package (https://github.com/deweylab/RSEM, accessed on 10 October 2024). All raw RNA sequencing data have been deposited in the National Center for Biotechnology Information (NCBI) under accession number PRJNA1238210.

### 4.4. GO and KEGG Functional Enrichment Analysis

DEGs were annotated using Gene Ontology (GO) and the Kyoto Encyclopedia of Genes and Genomes (KEGG). Enrichment analysis was conducted via the phyper function in R with Benjamini–Hochbergcorrection (FDR ≤ 0.05, https://github.com/deweylab/RSEM, accessed on 12 October 2024). Results were visualized using the Weishengxin Bioinformatics Platform (https://www.bioinformatics.com.cn/, accessed on 12 October 2024).

### 4.5. Identification of MIKC-Type AmMADS-box Genes

We downloaded the genome, GFF files, and protein files of snapdragon from the snapdragon Genome Database (http://bioinfo.sibs.ac.cn/Am/index.php, accessed on 18 October 2024). We retrieved the protein sequences of type-II AtMADS-box (Appendix A) from The *Arabidopsis* Information Resource (TAIR) database (https://www.arabidopsis.org/, accessed on 18 October 2024). In the initial screening phase, the BLAST algorithm is utilized. With the obtained AtMADS-box protein sequences as seed sequences, MIKC-type MADS-box proteins in snapdragon were preliminarily identified through local BLAST in TBtools (v0.6735) (E < 1 × 10^−10^).

Subsequently, we downloaded the Hidden Markov Model (HMM) configuration files for the SRF (M-type) domain (PF00319) and K-box (MIKC-type) domain (PF01486) from the Pfam database (http://pfam.xfam.org/, accessed on 19 October 2024). We used these HMMs to validate the initially identified AmMADS-box protein candidate sequences. We conducted a one-by-one identification and analysis of their conserved domains, and eliminated sequences with incomplete or redundant domains. Finally, the remaining validated sequences are determined as members of the AmMADS-box family (Appendix A).

### 4.6. Chromosomal Localization and Phylogenetic Analysis

The ‘Gene Location Visualize from GTF/GFF’ function in TBtools was used for the visual analysis of chromosomal localization and mapping the MADS-box genes onto their corresponding chromosomes [72]. To further explore the phylogenetic relationships between *Arabidopsis thaliana* and snapdragon MADS-box proteins, DNAMAN 9.0 was used to conduct a multiple amino acid sequence alignment of the obtained MIKC-type AtMADS-box and snapdragon AmMADS-box proteins. Subsequently, MEGA 11.0 (version 7.0) was employed to construct a phylogenetic tree based on the Maximum likelihood (ML) estimation method. The bootstrap value was set to 1000, and other parameters were set to their defaults [41]. Finally, the tree file was visualized and edited using iTOL (https://itol.embl.de/, accessed on 20 October 2024).

### 4.7. Protein Interaction and Subcellular Localization

Functional interaction analysis of the *Arabidopsis thaliana* homologous protein AmMADS was performed by constructing a protein-protein interaction (PPI) network using the STRING database (https://cn.string-db.org/, accessed on 3 November 2024) [37]. Subcellular localization was predicted using WoLF PSORT (https://wolfpsort.hgc.jp/, accessed on 5 November 2024).

### 4.8. Cis-Acting Element and TFBS Analysis

The Fasta Extract function in TBtools was utilized to obtain a 2000 bp upstream nucleotide sequence fragment (promoter sequence) from the snapdragon genome ffle for the AmMADS-box genes. This fragment was then submitted to the PlantCARE website (http://bioinformatics.psb.ugent.be/webtools/plantcare/html/, accessed on 5 November 2024) for the prediction and annotation of cis-acting elements in the gene promoters, facilitating functional classification and statistical analysis. Data visualization was conducted using TBtools and GraphPad PRISM (v9.3.1) Additionally, to further investigate transcriptional regulatory networks, the 2000 bp upstream sequences of AmMADS-box genes were analyzed using the Plant Transcription Regulation Map platform (https://plantregmap.gao-lab.org/index-chinese.php, accessed on 5 November 2024) for the prediction of potential transcription factor binding sites (TFBSs). Following the comprehensive integration of multi-source prediction data, the visualization of regulatory interactions was achieved through the built-in plotting utilities of TBtools.

### 4.9. WGCNA for Co-Expression Network Analysis

Based on the Weighted Gene Co-expression Network Analysis (WGCNA) system of the BMKCloud Platform (https://international.biocloud.net/zh/user/login, accessed on 15 November 2024), a gene co-expression network was constructed using the dynamic tree cut algorithm. Specifically, through the block-wise network construction strategy, the module merging similarity threshold (mergeCutHeight = 0.25) and the minimum module size (minModuleSize = 30) were set. Module Eigengenes (MEs) were calculated by eigengene centrality, and a multiple regression model between MEs and petal development phenotypes was established. Subsequently, core co-expression modules were identified through strict screening criteria. The network architecture was visualized in combination with the Topological Overlap Matrix (TOM), and the characteristics of the key biological pathways were analyzed through the module-trait interaction heatmap. 

### 4.10. Correlation Heatmap of ABCDE Model Genes, Phytohormones, and TFs Genes

To systematically investigate the regulatory interactions between ABCDE model genes and phytohormones/TFs, a global co-expression network was constructed. The phytohormone and TF datasets originated from the hub modules identified through WGCNA as described in Section 4.9. Spatial correlations between ABCDE model genes expression profiles and phytohormone/TF activity matrices were quantitatively evaluated using the Mantel test on the ChiPlot platform (https://www.chiplot.online, accessed on 17 November 2024), with 1000 permutations and Z-score normalization.

Correlation analysis between key factors and ABCDE model genes was performed through the online cloud platform (https://cloud.metware.cn, accessed on 17 November 2024). The correlation coefficient between key factors and ABCDE model genes that was screened for was the Pearson correlation coefficient (PCC), and a correlation was determined with a PCC > 0.8 or <−0.8 and a *p*-value < 0.05. Co-expression networks were visualized as an enrichment heatmap generated by TBtools.

Mantel tests (1000 permutations) were performed to assess correlations between ABCDE model genes expression and phytohormones/TFs (Z-score normalized). Significant correlations (*p* < 0.05, |PCC| > 0.8) were visualized as heatmaps using TBtools.

### 4.11. Nucleic Acid Isolation and qRT-PCR Analysis

Total RNA was isolated using the Eastep™ Super Total RNA Extraction Kit (Promega, Madison, WI, USA). First-strand cDNA synthesis was performed with 1 μg of total RNA in 20 μL reactions, using HiScript II Q RT SuperMix for qPCR (Vazyme, Nan jing, China) and oligo-dT (18)-MN primers following the manufacturer’s instructions. qRT-PCR was conducted with SYBR® Green Realtime PCR Master Mix-Plus (Takara, Tokyo, Japan) under the following conditions: polymerase activation for 30 s at 95 °C, followed by 40 cycles of 15 s at 95 °C, 15 s at 60 °C, and 25 s at 72 °C. The *AmUBI* (*Am03g01530*) gene served as an internal control, and gene expression was normalized to this reference gene. All primers used in these assays are listed in Appendix A, and each assay was carried out with three biological replicates. Data were visualized in GraphPad PRISM (v9.3.1).

### 4.12. VIGS Vector Construction and Transient Transformation

The pTRV1/pTRV2 binary vectors and *Agrobacterium tumefaciens* GV3101 strain were obtained from Beijing Huayueyang Biotech. Co., Ltd. Gene-specific primers targeting the conserved ORF region of *A. majus AmMADS85* amplified a ~300 bp fragment via PCR (Appendix A). After double digestion with *EcoR*I/*BamH*I and gel purification, the fragment was ligated into pTRV2. The recombinant plasmid pTRV2-*AmMADS85* was transformed into GV3101 cells, verified by colony PCR, and cultured in LB medium with 50 mg/L kanamycin.

For infiltration, *Agrobacterium* cultures (OD₆₀₀ = 0.8–1.0) carrying pTRV1 and pTRV2-*AmMADS85* (or empty pTRV2 as control) were mixed 1:1 in MMA buffer (10 mM MES, pH 5.6; 10 mM MgCl_2_; 100 μM AS). Three flowers per plant (9 plants/group) were syringe-infiltrated into petals. Post-inoculation, plants were kept in darkness for 3 h. Samples were collected at 0, 6, 12, and 18 h, flash-frozen in liquid nitrogen, and stored at −80 °C. Gene silencing efficiency was analyzed by qRT-PCR (Section 4.11). All experiments included three biological replicates.

### 4.13. Statistical Analysis

Data were analyzed using three biological replicates (n = 3). DEGs in RNA-seq were identified via edgeR (v3.36.0) in R (v4.2.3) with FDR ≤ 0.05 and |log_2_(fold change)| ≥ 0.5. Gene expression correlation was assessed by the Mantel test (Bray–Curtis distance, 999 permutations) using a vegan version (v2.6-4). Visualization was performed in GraphPad PRISM (v9.3.1). Statistical significance was defined as *p* < 0.05, with error bars indicating mean ± SD.

## 5. Conclusions

Through multiple integration approaches (phenotypic analysis, RNA-seq, WGCNA, PPI, and VIGS validation), this study delineated the molecular regulatory network driving petal malformation in *A. majus* AmDP2. A potential combinatorial regulatory framework emerged at three levels (Figure 8): (1) *SEP3* (*AmMADS25/61*) and *SEP2* (*AmMADS85*) exhibited opposing expression patterns. *SEP3* drove malformation via interactions with B-class *AP3* (*AmMADS37/3*) and A-class *AP1* (*AmMADS33/43*), while SEP2 maintained normal development. (2) *SEP3 (AmMADS25/61)* associates with *AGL5 (AmMADS16)* in a co-expression network linked to floral organogenesis, supported by PPI and VIGS evidence. (3) *AGL5* negatively correlated with auxin-responsive genes (*SAUR1/IAA13*), impairing auxin signaling. These findings preliminary establish a “genes–TFs–hormones” tripartite model where E-class MADS-box *SEP2/SEP3* genes integrate developmental and hormonal cues. Our work provides CRISPR-targetable hubs (*SEP3/SEP2/AGL5*) for ornamental crop improvement and advances our understanding of floral evolution through transcriptomic network biology.

## Figures and Tables

**Figure 1 ijms-26-04450-f001:**
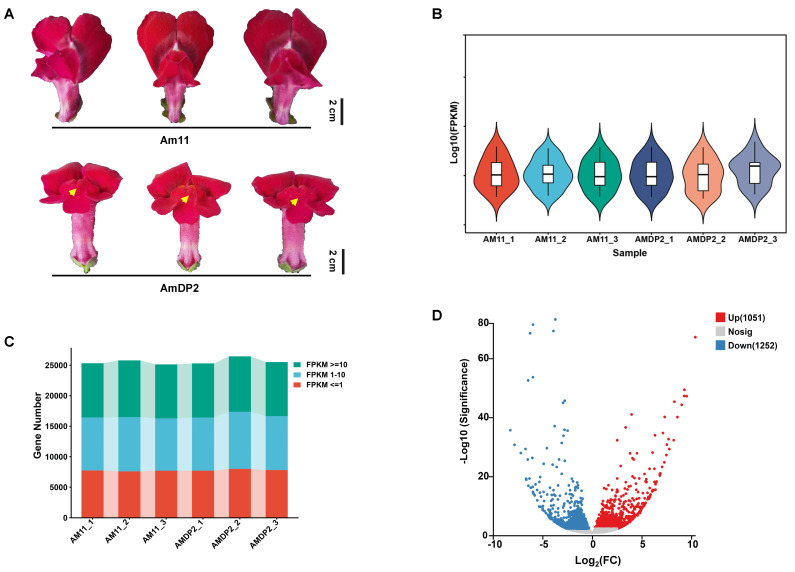
Phenotypic and transcriptomic analysis of petal malformation in snapdragon. (**A**) Phenotypic comparison of normal petals (Am11) and malformed petals (AmDP2). Scale bar: 2 cm. (**B**) RNA-seq total normalized expression per replicate. (**C**) Distribution of gene counts in different FPKM intervals for each sample. (**D**) Volcano plot of differentially expressed genes (DEGs) between Am11 and AmDP2. Each dot represents a gene: red (upregulated), blue (downregulated), gray (non-significant).

**Figure 2 ijms-26-04450-f002:**
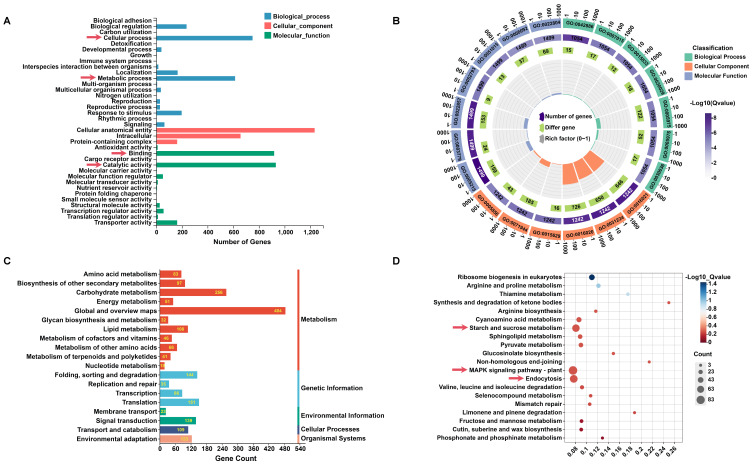
GO and KEGG pathway analysis of DEGs. (**A**) Functional annotation distribution of DEGs using Gene Ontology (GO). *X*-axis: number of annotated genes; *Y*-axis: GO categories. The red arrows indicate the entries with the largest number of annotated genes in the GO classifications (CC, BP, MF). (**B**) Circular plot of the top 20 enriched GO terms. Layers: (1) top 20 GO terms, (2) genome-wide background counts and enrichment *p*-values, (3) enriched gene counts, (4) enrichment factors. Colors: biological process (BP—green), cellular component (CC—orange-red), molecular function (MF—purple). (**C**) Classification of DEGs into KEGG pathways grouped into five categories: metabolism, genetic information processing, environmental information processing, cellular processes, organismal systems. (**D**) Bubble chart of the top 20 enriched KEGG pathways. *X*-axis: enrichment ratio; *Y*-axis: pathway name; bubble size: number of genes; color: enrichment *Q*-value (darker = lower *Q*-value). The red arrows indicate the key pathways with significant enrichment.

**Figure 3 ijms-26-04450-f003:**
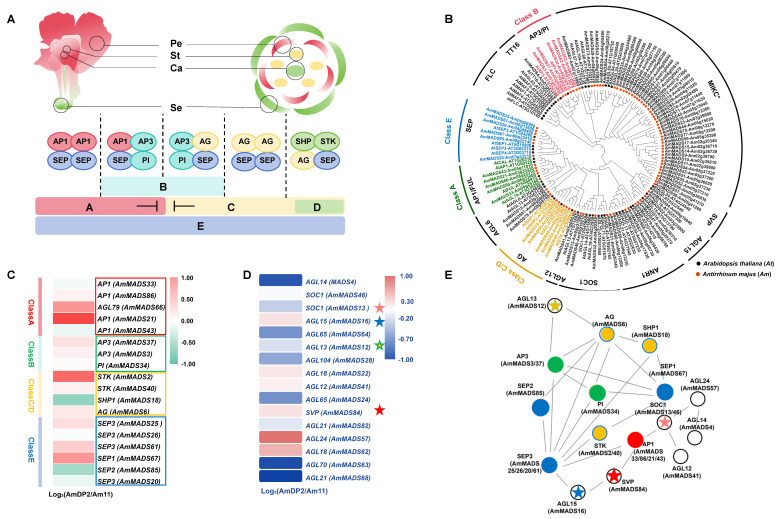
Integrated analysis of *AmMADS* genes associated with petal malformation. (**A**) ABCDE floral organ development model. (**B**) Maximum likelihood (ML) phylogenetic tree of MIKC-type MADS-box genes in *A. majus* (Am) and *A. thaliana* (At). Red circles: AmMADS proteins; Black circles: AtMADS proteins. Colored labels: ABCDE model genes. (**C**,**D**) RNA-seq-based differential expression of *AmMADS* genes in petals of Am11 (control) and AmDP2 (malformed). Values represent normalized log2 FPKM. (**E**) Protein-protein interaction network of ABCDE model proteins in *A. majus*, inferred from *A. thaliana* data. Red nodes indicate class A genes, green nodes indicate class B genes, yellow nodes indicate class C/D genes, blue nodes indicate class E genes, and nodes without fill color represent MTFs, where the pentagrams represent key TFs. Edges represent interactions between genes. The red, green, and yellow-green interlaced five-pointed stars represent the MTFs interacting with Class A, Class B, and Class B + C/D, respectively.

**Figure 4 ijms-26-04450-f004:**
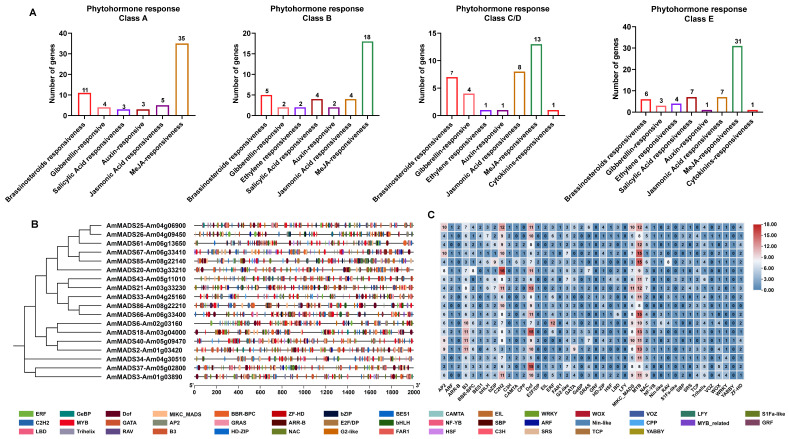
Hormone response element and transcription factor (TF) binding site analysis of promoter regulatory elements in differentially expressed ABCDE model genes. (**A**) Phytohormone-responsive element enrichment. (**B**) Distribution of TFBS in promoter regions. (**C**) Number of TFs binding sites as shown in (**B**).

**Figure 5 ijms-26-04450-f005:**
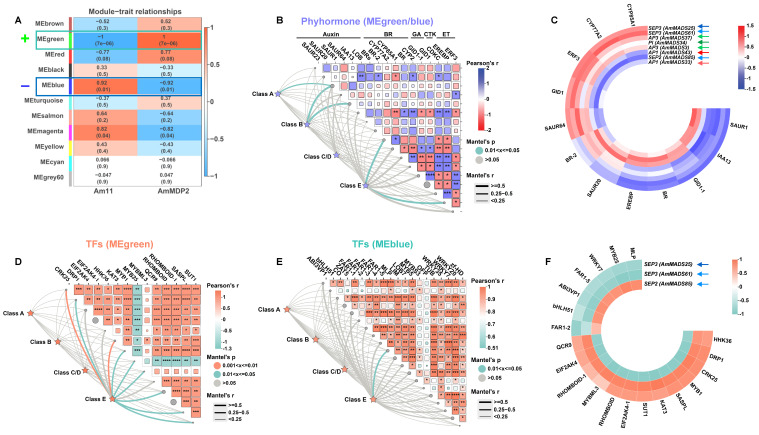
Module–trait associations and statistical validations. (**A**) Correlation matrix between module DEGs (rows) and malformed petal traits (columns). Each row corresponds to a module characteristic gene, and each column corresponds to a trait. Each cell contains a corresponding correlation and *p*-value. According to the color legend, the table is color-coded by correlation. (**B**) Mantel test correlations between differentially expressed ABCDE model genes and phytohormones in MEgreen and MEblue modules. (**C**) Heatmaps of significant correlations between key differentially expressed A-class, B-class, and E-class genes as shown in (**B**) and phytohormones. (**D**,**E**) Mantel test correlations between differentially expressed ABCDE model genes and TFs in MEgreen and MEblue modules. (**F**) Heatmaps of significant correlations between key differentially expressed E-class genes as shown in (**D**,**E**) and TFs. Correlation strength: red lines represent strong correlations (*p* < 0.001), green lines represent moderate correlations (*p* < 0.01), and gray lines represent no significant correlation (*p* ≥ 0.05). *, **, ***, and **** indicate significance at *p* < 0.05, *p* < 0.01, *p* < 0.001, and *p* < 0.0001. The colors of the arrows correspond to different functional gene class: red (A-class), green (B-class), and blue (E-class).

**Figure 6 ijms-26-04450-f006:**
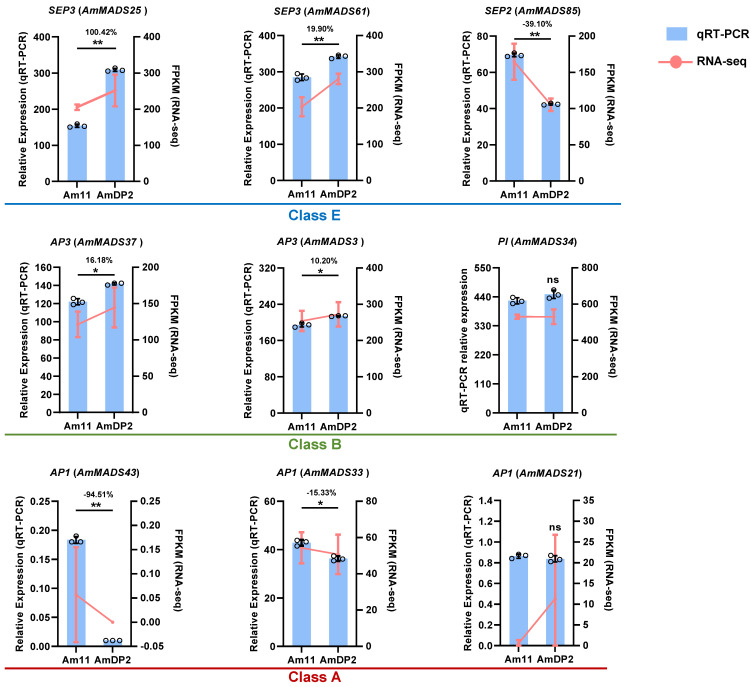
Validation of candidate ABCDE model genes in Am11 and AmDP2, normalized to *AmUBI* (internal control). The expression levels obtained by RNA-seq and qRT-PCR are shown with a line chart and histogram, respectively. Bars represent means ± SD (*n* = 3 biological replicates, with individual data points shown). * and ** indicate significant differences between Am11 and AmDP2 (*t*-test, *p* < 0.05 or *p* < 0.01, *n* = 3), and ns denotes not significant.

**Figure 7 ijms-26-04450-f007:**
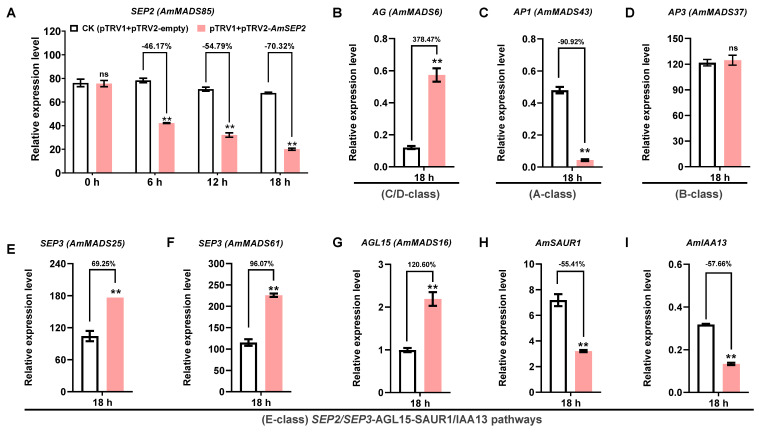
Silencing of *AmMADS85* in *A. majus* Am11 petals by VIGS system. (**A**) qRT-PCR confirmed *AmMADS85* gene silencing in pTRV2-*AmSEP2*-inoculated petals at 6, 12, and 18 h post-inoculation (hpi). Mock-treated (agro-infiltrated with pTRV1 + pTRV2-empty); silenced (agro-infiltrated with pTRV1 + pTRV2-*AmMADS85*). (**B**–**I**) The expression levels of C/D-class *AG* (*AmMADS6*), A-class *AP1* (*AmMADS43*), B-class *AP3* (*AmMADS37*), E-class *SEP3* (*AmMADS25/61*), co-regulated MADS factor *AGL5* (*AmMADS16*), and hormone-responsive genes *AmSAUR1/AmIAA13*. Bars represent means ± SD (*n* = 3 biological replicates, with individual data points shown). ** indicates significant differences between CK and silencing petals (*t*-test, *p* < 0.01, *n* = 3); ns denotes not significant.

**Figure 8 ijms-26-04450-f008:**
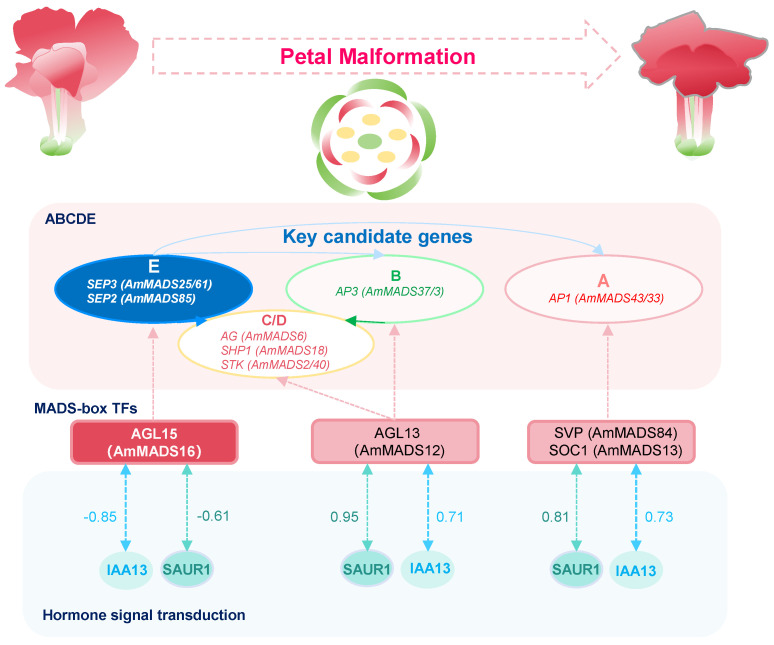
Hypothetical gene network underlying petal malformation in *A. majus*. This schematic illustrates a core gene network involving E-class AmMADS-box genes (*AmMADS25/61/85*) interacting with C/D/B/A-class MADS-box genes. The network is associated with petal malformation through IAA13- and SAUR1-mediated auxin signaling pathways. Potential coordination occurs between these genes and MTFs (e.g., *AmMADS16*) in modulating developmental gene expression patterns linked to floral organogenesis.

## Data Availability

The datasets supporting the results presented in this manuscript are included within the article (and its Appendix A).

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
