# Peer review of "Transcriptome Analysis Reveals Association of E-Class AmMADS-Box Genes with Petal Malformation in Antirrhinum majus L."

_ijms, 2025, doi:10.3390/ijms26094450_

Round 1

Reviewer 1 Report

Comments and Suggestions for Authors

Summary
The authors present an insightful evaluation of MADS-BOX gene expression in Antirrhinum majus, contributing to a better understanding of floral development, particularly the genetic mechanisms underlying petal malformation. The manuscript will be accepted after addressing minor revisions.

Title
Ensure the species name is italicized.

Abstract

  • Line 32: Specify the context of "key target."

Introduction

  • Consider moving lines 64–74 before describing the ABCE model to first provide a clearer overview of MADS-BOX genes and their role in floral architecture.
  • MADS-BOX sequences could serve as molecular markers for marker-assisted selection (MAS). Reinforce this point by incorporating relevant references: https://doi.org/10.3390/agriculture11070622; https://doi.org/10.1007/s11105-013-0597-9; https://doi.org/10.3390/horticulturae91213109

Results

  • Lines 93–95 and 97–99 contain methodological details that belong in the Materials and Methods section.
  • Figure 1B: Clarify that this violin plot represents total normalized expression per replicate rather than an expression profile.
  • Figure 3E: Specify that the network plot was adapted from the STRING database in the figure legend. Explain the meaning of different symbols/colors used.

Discussion

  • The discussion is well-developed and supports the statement in line 149 regarding MADS-BOX transcription factors (TFs) and their involvement in primary and secondary metabolism.
  • Consider adding a brief paragraph on the potential correlation between MADS-BOX TFs and secondary metabolite modulation in A. majus—an aspect not tested in this study but valuable for future research.

Materials and Methods

  • Include additional details about plant experimental conditions.

Author Response

Reviewer 1:

The authors present an insightful evaluation of MADS-BOX gene expression in Antirrhinum majus, contributing to a better understanding of floral development, particularly the genetic mechanisms underlying petal malformation. The manuscript will be accepted after addressing minor revisions.

Answer (A): Thanks for your encouraging comments. We have incorporated the suggested additions into the “Discussion” sections, explicitly addressing the potential correlation between MADS-box TFs and secondary metabolite modulation, as well as their utility as molecular markers for MAS. Specifically, the “Abstract”, “Discussion”, and “Material and Methods” sections and partial Figure legends have been appropriately modified to offer a more comprehensive exposition of this study.

  1. Ensure the species name is italicized.

A: Thank you for your suggestions. We have revised the manuscript to ensure that all species names are in italics.

  1. Line 32: Specify the context of "key target."

A: We have made appropriate revisions in the “Abstract” and “​Conclusions” sections. Please see lines 30-35 and lines 676-678.

  1. Consider moving lines 64-74 before describing the ABCE model to first provide a clearer overview of MADS-BOX genes and their role in floral architecture.

A: We appreciate your suggestion. This may also be a good way to explain, but after careful consideration, we still chose to first introduce the ABCDE model as a foundational framework to anchor readers’ understanding, followed by functional details.

  1. MADS-BOX sequences could serve as molecular markers for marker-assisted selection (MAS). Reinforce this point by incorporating relevant references: https://doi.org/10.3390/agriculture11070622; https://doi.org/10.1007/s11105-013-0597-9; https://doi.org/10.3390/horticulturae91213109.

A: Thank you for your valuable suggestion. Thank you for your valuable suggestion to strengthen the discussion on MADS-box genes as molecular markers for marker-assisted selection (MAS). We have made appropriate revisions in the “Discussion” sections. Please see lines 466-473.

   While the metioned reference (https://doi.org/10.3390/agriculture11070622; https://doi.org/10.1007/s11105-013-0597-9;https://doi.org/10.3390/horticulturae91213109.) could not be found, we have supplemented the discussion with two other refs that validate the utility of MADS-box markers in plant improvement:

  • ​​Zong, W.B.; Song, Y.A.; Xiao, D.D.; Guo, X.T.; Li, F.Q.; Sun, K.L.; Tang, W.J.; Xie, W.H.; Luo, Y.; Liang, S.; et al. Dominance complementation of parental heading date alleles of Hd1, Ghd7, DTH8, and PRR37 confers transgressive late maturation in hybrid rice. Plant J. 2024, 118, 2108–
  • Zuriaga, E.; Romero, C.; Blanca, J.M.; Badenes, M.L. Resistance to Plum Pox Virus (PPV) in apricot (Prunus armeniaca) is associated with down-regulation of two MATHd genes. BMC Plant Biol. 2018, 18, 25.

  1. Lines 93–95 and 97–99 contain methodological details that belong in the Materials and Methods section.

A: Thanks. We have made appropriate revisions in the “Results” and “Material and Methods” sections. Please see lines 88-96, lines 495-497, lines 503-504 and lines 519-521.

  1. Figure 1B: Clarify that this violin plot represents total normalized expression per replicate rather than an expression profile.

A: Thanks. According to your suggestion, we have clearly stated in legend of Figure 1 B that the violin plot shows the total normalized expression of per replicate.

Figure 3E: Specify that the network plot was adapted from the STRING database in the figure legend. Explain the meaning of different symbols/colors used.

A: Thanks. We have added information in the legend of Figure 3E: Nodes represent genes. Red nodes indicate class A genes, green nodes indicate class B genes, yellow nodes indicate class C/D genes, blue nodes indicate class E genes, and nodes without fill color represent MTFs, where the pentagrams represent key transcription factors. Edges represent interactions between genes.

  1. The discussion is well-developed and supports the statement in line 149 regarding MADS-BOX transcription factors (TFs) and their involvement in primary and secondary metabolism.

Consider adding a brief paragraph on the potential correlation between MADS-BOX TFs and secondary metabolite modulation in A. majus—an aspect not tested in this study but valuable for future research.

A: Thank you for your unique and detailed insights. We have made supplementary additions in the “Discussion” section, and the specific content is as follows. Please see lines 474-490.

While our study focuses on MADS-box TFs in primary metabolism (e.g., starch/sucrose metabolism, Figure 2D), emerging evidence suggests their potential role in secondary metabolite regulation—a largely unexplored frontier in A. majus. KEGG pathway analysis revealed that SVP (Am08g16940) and SOCI (Am05g19210/Am02g37900) are enriched in metabolic and cellular processes, hinting at possible crosstalk with secondary metabolite biosynthesis. For instance, overexpression of CsMADS3 in citrus and tomato enhances carotenoid biosynthesis while upregulating chlorophyll degradation genes (Zhu et al., 2021), paralleling potential Antirrhinum MADS-box regulation of terpenoid/phenylpropanoid pathways via downstream enzymes (e.g., CYP76B6) (Qiao et al., 2023). Notably, the MAPK signaling pathway (KEGG: ko04016), significantly enriched here, is tightly linked to secondary metabolism. MAPK cascades regulate JA biosynthesis, a hormone critical for defense-related secondary metabolite accumulation (Lin L et al., 2024). This hypothesis is further supported by JA-responsive elements in E-class MADS-box genes (Figure 4A). Future studies integrating metabolomics and ChIP-seq could clarify whether A. majus MADS-box TFs directly govern secondary metabolite pathways, expanding their functional scope in floral development.

  1. Include additional details about plant experimental conditions.

A: Thanks. We have added it in the “Materials and Methods” section. Please see lines 495-497.

Submission Date

26 March 2025

Date of this review

02 Apr 2025 10:00:16

A: Thanks again. We have incorporated your suggestions and conducted a thorough revision of the entire text.

Reviewer 2 Report

Comments and Suggestions for Authors

I go through the manuscript entitled Transcriptome Analysis Reveals E-Class AmMADS-Box Genes Drive Petal Malformation in Antirrhinum majus L.. I find it very interesting. However it require serious revision bacause it lacks scientific writeup and data representation. Some data do not support the conclusion meanwhile the manuscript is also drafted in very over complex way. Firstly I suggest remove the word like drives from title because he word "drive" implies causation, which may overstate the findings without experimental validation such as knockout/overexpression. It may be regulate or use other neutral terms like role in to reflect correlative evidence.

The abstract is overly complex and lacks clarity. Phrases like "multi-dimensional integration approaches," "molecular scaffolds," and "hierarchical interaction networks" are jargon-heavy and undefined. The abstract does not clearly state the hypothesis, methodology, or specific findings. Phrases like "novel preliminary model" and "actionable resources for CRISPR-based improvement" are speculative without experimental validation. Abstract need to be scientific, simple and to the point. For example, Petal malformation in Antirrhinum majusimpacts ornamental value, but its genetic basis remains poorly understood. We compared transcriptomes of the wild-type (Am11) and a petal-malformed mutant (AmDP2) to identify 2,303 differentially expressed genes (DEGs), including E-class MADS-box genes (SEP3, SEP2, SEP1). Co-expression analysis revealed interactions between SEP3 and B/A-class MADS-box genes (AP3, PI, AP1), as well as auxin signaling genes (SAUR1, IAA13). qRT-PCR validated upregulation of SEP3 and downregulation of SEP2 in AmDP2. Our results suggest that E-class MADS-box genes regulate petal malformation via crosstalk with hormonal pathways. These findings provide candidate targets for further functional studies in snapdragon. Keywords need to be ascending alphabetical order such as ABCDE model, hormone crosstalk, MIKC-type MADS-box genes, petal malformation, RNA-seq, snapdragon (Antirrhinum majus), weighted gene co-expression network

In introduction section italicize gene names (e.g., SVP, SOC1, KNOX, bHLH) and protein names should be kept straight throughourt the manuscript. Also see The A-class genes (AP1, AP2, FUL) + E-class genes (SEP1-4). In introduction section the rationale for focusing on AmDP2 is unclear. Explain why this mutant was chosen over others. Simplify sentences like "floral morphogenesis is driven by multidimensional regulatory networks" to "floral morphogenesis involves hormonal and transcriptional networks." In Figure 6 Correct Y-axis labels to "Relative Expression (qRT-PCR)" (left) and "FPKM (RNA-seq)" (right). Also its legends should begin with word Validation of DEGs... instead of Expression levels. Replace low-resolution figures with high-resolution versions (minimum 300 dpi). Move Table 1 to Supplementary Materials. The conclusion is verbose and repetitive. Focus on key findings such as SEP3 upregulation correlates with malformation. Limitations such as lack of functional validation. Future directions such as CRISPR editing of SEP3/SEP2. Clarify how this work advances the field beyond existing ABCDE model studies. In materials and methods add a Statistical Analysis subsection. Describe tests used such as t-tests for qRT-PCR, FDR thresholds for RNA-seq. Specify software such as edgeR, PRISM for DEGs, Mantel tests for correlations.

I would also like to ask the coauthors why was AmDP2 chosen? Are there other mutants with similar phenotypes? What evidence supports SEP2’s role in "maintaining normal petal development"? How will you test the proposed SEP3-AGL15-SAUR1/IAA13 network such as Y2H, CRISPR? How does auxin/GA crosstalk specifically influence petal morphology?

I see many mistakes in references section please double cross check the references for correction of typos. Also, compare findings with recent work on SEP genes in Petunia or Arabidopsis.

Overall, the work is well-conducted; however, substantial and significant revisions are necessary for the manuscript to meet the standards required for publication. I recommend accepting the manuscript pending minor revisions, provided the authors adequately address the suggestions outlined above.

Comments on the Quality of English Language

The manuscript needs to be thoroughly rephrased and reviewed by a native English speaker prior to submitting the revised version.

Author Response

REVIEWER COMMENTS:

Reviewer 2:

I go through the manuscript entitled Transcriptome Analysis Reveals E-Class AmMADS-Box Genes Drive Petal Malformation in Antirrhinum majus L.. I find it very interesting. However it require serious revision bacause it lacks scientific writeup and data representation. Some data do not support the conclusion meanwhile the manuscript is also drafted in very over complex way.

Answer (A): Thank you for your constructive feedback. To address concerns regarding SEP2’s role in petal development, we supplemented experimental evidence by transiently silencing SEP2 (AmMADS85) in wild-type A. majus Am11 petals via VIGS. The data predicts that SEP2 silencing may exacerbates malformation primarily through activation of the E-class SEP2/SEP3-AGL15-SAUR1/IAA13 regulatory axis, directly linking SEP2 dysfunction to petal developmental defects. Moreover, clarify how this work makes advancements in the field beyond the existing ABCDE model studies. We carefully revised the entire manuscript and rewrote the abstract and conclusion to make them more concise and focused. A foreign expert in the relevant field rigorously polished the English language of the manuscript. We also modified the figures to make them clearer. Specifically, the “Abstract”, “Results”, “Discussion”, “Material and Methods” and “Conclusion” sections and partial Figure legends and Figure 3, 6, 7, 8 have been appropriately modified to offer a more comprehensive exposition of this study.

  1. Firstly, I suggest remove the word like drives from title because he word "drive" implies causation, which may overstate the findings without experimental validation such as knockout/overexpression. It may be regulate or use other neutral terms like role in to reflect correlative evidence.

A: Thank you for your suggestion. We revised title as “Transcriptome Analysis Reveals E-Class AmMADS-Box Genes Regulate Petal Malformation in Antirrhinum majus L.”

  1. The abstract is overly complex and lacks clarity. Phrases like "multi-dimensional integration approaches," "molecular scaffolds," and "hierarchical interaction networks" are jargon-heavy and undefined. The abstract does not clearly state the hypothesis, methodology, or specific findings. Phrases like "novel preliminary model" and "actionable resources for CRISPR-based improvement" are speculative without experimental validation. Abstract need to be scientific, simple and to the point. For example, Petal malformation in Antirrhinum majus impacts ornamental value, but its genetic basis remains poorly understood. We compared transcriptomes of the wild-type (Am11) and a petal-malformed mutant (AmDP2) to identify 2,303 differentially expressed genes (DEGs), including E-class MADS-box genes (SEP3, SEP2, SEP1). Co-expression analysis revealed interactions between SEP3 and B/A-class MADS-box genes (AP3, PI, AP1), as well as auxin signaling genes (SAUR1, IAA13). qRT-PCR validated upregulation of SEP3 and downregulation of SEP2 in AmDP2. Our results suggest that E-class MADS-box genes regulate petal malformation via crosstalk with hormonal pathways. These findings provide candidate targets for further functional studies in snapdragon.

A: Thank you for your suggestion. We rewrote the abstract and according text.

  1. Keywords need to be ascending alphabetical order such as ABCDE model, hormone crosstalk, MIKC-type MADS-box genes, petal malformation, RNA-seq, snapdragon (Antirrhinum majus), weighted gene co-expression network.

A: Thanks. We revised it.

  1. In introduction section italicize gene names (e.g., SVP, SOC1, KNOX, bHLH) and protein names should be kept straight throughourt the manuscript. Also see the A-class genes (AP1, AP2, FUL) + E-class genes (SEP1-4).

A: We sincerely appreciate your suggestions and revised the whole text.

  1. Simplify sentences like "floral morphogenesis is driven by multidimensional regulatory networks" to "floral morphogenesis involves hormonal and transcriptional networks."

A: Thanks. We have revised the text.

  1. In Figure 6 Correct Y-axis labels to "Relative Expression (qRT-PCR)" (left) and "FPKM (RNA-seq)" (right). Also its legends should begin with word Validation of DEGs... instead of Expression levels.

A: Thanks. We have corrected the Y-axis labels of Figure 6 as “Relative Expression (qRT-PCR)” (left axis) and “FPKM (RNA-seq)” (right axis) as suggested, and the legend has been appropriately modified.

  1. Replace low-resolution figures with high-resolution versions (minimum 300 dpi).

A: Thanks. We revised all the Figures.

  1. Move Table 1 to Supplementary Materials.

A: Thanks. We revised it. Please see Table S1 in the revised manuscript.

  1. The conclusion is verbose and repetitive. Focus on key findings such as SEP3 upregulation correlates with malformation.

A: Thank you for your valuable suggestion. We have revised the conclusion to prioritize key findings, explicitly emphasizing the correlation between SEP3 upregulation and petal malformation. Please see lines 6763-678.

  1. In introduction section the rationale for focusing on AmDP2 is unclear. Explain why this mutant was chosen over others.

I would also like to ask the coauthors why was AmDP2 chosen? Are there other mutants with similar phenotypes?

A: Thank you for raising this important point. We added a brief description in the “Introduction” section. Please see lines 74-76.

We selected the AmDP2 mutant for three key reasons, detailed below:

 (1) Unique Phenotype​​: AmDP2 exhibits distinct petal malformations (fused petals, disrupted boundaries; Figure 1A), mimicking ABCDE model disruptions. Unlike def/squa mutants (B-class gene defects), AmDP2 uniquely highlights E-class SEP2/SEP3 dysregulation.

  • E-class Gene Imbalance​​: AmDP2 shows SEP2 downregulation and SEP3 upregulation, contrasting with B/C-class mutants (e.g., def). VIGS data confirm SEP2 loss drives aberrant C/D-class gene activation (e.g., AG; Figure 7), a mechanism unexplored in prior studies.
  • Hormonal Relevance​​: AmDP2 exhibits altered auxin signaling (SAUR1/IAA13), central to petal morphogenesis. Unlike classical ABCDE studies, we aimed to resolve how SEP genes integrate hormonal signals to maintain floral homeostasis.

​

  1. What evidence supports SEP2’s role in "maintaining normal petal development"? How will you test the proposed SEP3-AGL15-SAUR1/IAA13 network such as Y2H, CRISPR? How does auxin/GA crosstalk specifically influence petal morphology?

Limitations such as lack of functional validation. Future directions such as CRISPR editing of SEP3/SEP2.

A: I fully concur with this crucial point. To promptly provide some evidence supporting SEP2’s role in “maintaining normal petal development”, we conducted an experiment in which AmMADS85 (SEP2, E-class) was transiently silenced in the petals of wild-type Antirrhinum majus Am11.

Our data show that SEP2 is downregulated in AmDP2 (Figure 6A), and its silencing via VIGS may exacerbates malformation primarily through activation of the ​​E-class SEP2/SEP3-AGL15-SAUR1/IAA13 regulatory axis (Figure 7).

Please see “2.7. Silencing of AmSEP2 Promotes Petal Malformation in A. majus: Gene Network Interactions​” in the “Results” section (lines 234-263), “4.12. VIGS Vector Construction and Transient Transformation in the “Material and Methods” section  (lines 641-662), Figure 7 and the relevant descriptions throughout the manuscript。

Since stable genetic and biochemical validations have not been carried out, there are limitations to the points proposed in this study. Therefore, in-depth experiments such as validation of the SEP3-AGL15-SAUR1/IAA13 network, Yeast Two-Hybrid (Y2H)/Co-Immunoprecipitation (Co - IP), CRISPR editing, and hormone profiling are necessary in the future and will be the focus of our subsequent research (lines 487-490).

  1. Clarify how this work advances the field beyond existing ABCDE model studies.

A: We thank the reviewer for recognizing the novelty of our findings. Our work advances beyond classical ABCDE models by demonstrating that SEP2/SEP3 act as dynamic regulators integrating hormonal (auxin) and transcriptional co-regulator (AGL15) signals to refine petal patterning. Crucially, we ​​initially uncovered​​ how the SEP3-AGL15-SAUR1/IAA13 axis disrupts auxin homeostasis—a mechanism unexplored in classical ABCDE frameworks. This expands the regulatory paradigm to incorporate hormonal feedback loops, offering a ​​systems-level perspective​​ on floral morphogenesis (Figure 8).

  1. In materials and methods add a Statistical Analysis subsection. Describe tests used such as t-tests for qRT-PCR, FDR thresholds for RNA-seq. Specify software such as edgeR, PRISM for DEGs, Mantel tests for correlations.

A: Thanks. We have added “4.13 Statistical Analysis” in the “Materials and Methods” section. Please see lines 656-662.

  1. I see many mistakes in references section please double cross check the references for correction of typos. Also, compare findings with recent work on SEP genes in Petunia or Arabidopsis.

A: Sorry for the mistakes, we revised all the references.

Additionally, according to your suggestions, we have made changes in the “Discussion” section. Please see lines 348-371.

Overall, the work is well-conducted; however, substantial and significant revisions are necessary for the manuscript to meet the standards required for publication. I recommend accepting the manuscript pending minor revisions, provided the authors adequately address the suggestions outlined above.

A: We sincerely appreciate your kindly comments and constructive suggestions. We have added some relevant validation experiments and carefully revised the whole manuscript. Hope to meet the standards required for publication.

Comments on the Quality of English Language

The manuscript needs to be thoroughly rephrased and reviewed by a native English speaker prior to submitting the revised version.

A: Following our revision, foreign expert in related fields rigorously revised the manuscript’s English language.

Submission Date

26 March 2025

Date of this review

04 Apr 2025 21:53:05

A: Thanks again. We have incorporated your suggestions and conducted a thorough revision of the entire text.